# LLM-DAMVC: A Large Language Model Assisted Dynamic Agent for Multi-View Clustering

**Haiming Xu[1]**    **Qianqian Wang[1]**[*]

[1]School of Telecommunications Engineering, Xidian University, Xi'an, China
24011211044@stu.xidian.edu.cn, qqwang@xidian.edu.cn

## Abstract

Multi-view clustering integrates the consistency and complementarity of different views to achieve unsupervised data grouping. Existing multi-view clustering methods primarily confront two challenges: i) they generally perform feature extraction in the feature domain, which is sensitive to noise and may neglect cluster-specific information that is indistinguishable in the original space; ii) current dynamic fusion methods adopt static strategies to learn weights, lacking capability to adjust strategies adaptively under complex scenarios according to variations in data distribution and view quality. To address these issues, we propose a large language model assisted dynamic agent for multi-view clustering (LLM-DAMVC), a novel framework that recasts multi-view clustering as a dynamic decision-making problem orchestrated by a large language model. Specifically, each view is equipped with complementary agents dedicated to feature extraction. A dual-domain contrastive module is introduced to optimize feature consistency and enhance cluster separability in both the feature domain and frequency domain. Additionally, an LLM-assisted view fusion mechanism provides a flexible fusion weight learning strategy that can be adaptively applied to complex scenarios and significantly different views. Extensive experimental results validate the effectiveness and superiority of the proposed method.

## 1 Introduction

Nowadays, data from different sources or modalities (*i.e.*, multi-view data) have become increasingly ubiquitous, such as user profiles and behavioral data in social networks, multi-modal scanning results in medical imaging, and multi-variate sensor data generated by IoT devices (1; 2; 3; 4; 5; 6). These multi-view data often contain richer and more comprehensive information than single-view data but also pose challenges such as heterogeneity, noise interference, and missing labels. Multi-View Clustering (MVC), which aims to effectively partition unlabeled data by leveraging the consistent and complementary information across views, has become a core technology in multi-view analysis, data mining, and pattern recognition (7; 8; 9; 10).

To effectively fuse multi-view information, researchers have proposed various MVC methods. Early approaches primarily relied on non-negative matrix factorization, subspace learning, or spectral clustering (11; 12; 13; 9; 14; 15). With the advancement of deep learning, deep MVC methods (e.g., those based on autoencoders, graph neural networks, or contrastive learning) have made significant progress in representation learning and exploring nonlinear relationships (16; 17; 18; 19; 20; 21). Notably, some studies have attempted to address view fusion by using attention mechanisms to adaptively aggregate view features (22; 23). Additionally, to learn a consistent representation with discriminative information, contrastive learning has been widely adopted to improve clustering performance by maximizing the similarity between positive samples and meanwhile minimizing that

---

[*]Corresponding Author

39th Conference on Neural Information Processing Systems (NeurIPS 2025).

between negative samples (24; 25). Despite their effectiveness, most of them focus on feature-domain consistent representation learning, which neglects imperceptible and indistinguishable features in the original feature space. Nevertheless, these features can be separable easily in the frequency domain. Moreover, these methods typically rely on predefined rules to assign weights for each view, making them inflexible when applied in complex scenarios and processing heterogeneous views. Especially, their static strategies for determining view weights may lack adaptability in adjusting strategies when facing drastic variations in view quality at the sample level. Existing methods generally lack a centralized coordination mechanism capable of understanding view content, assessing view quality, and performing intelligent decision-making.

To address these challenges and inspired by the powerful decision-making capacity of Large-scale Language Models (LLMs), we propose an LLM-Assisted Dynamic Agent for Multi-View Clustering (LLM-DAMVC). Different from existing works that introduce LLMs to MVC for feature extraction (26), LLM-DAMVC leverages the semantic reasoning and decision-making capabilities of LLMs for adaptive view fusion in MVC. This is achieved through a synergistic architecture incorporating multiple agents per view and a dual-domain contrastive learning mechanism. Specifically, each view is equipped with a standard agent and an adversarial agent, respectively focusing on semantic feature extraction and adversarial learning. A dual-domain contrastive learning module is introduced to optimize feature consistency in the feature domain and frequency domain. Finally, we utilize the information of the semantic feature quality, cluster structure quality, and intra-cluster compactness to construct a prompt for LLM to dynamically learn a fusion weight for each view. The main contributions of LLM-DAMVC are summarized as follows:

- We propose a novel MVC framework named LLM-DAMVC by introducing LLM as a decision-making module to evaluate the view quality in real time and dynamically assign aggregation weights, so that the model can adaptively adjust the fusion strategy according to the view characteristics and sample distribution.

- We build a dual-domain contrastive learning module that integrates frequency-domain contrastive learning with feature-domain contrastive learning. It maps the features to the frequency space through the FFT transform, capturing imperceptible and indistinguishable features in the original feature space and significantly enhancing the feature representation capability.

- We conduct extensive experiments on several benchmark datasets and compare our method with state-of-the-art MVC methods. The experimental results and analyses demonstrate the effectiveness of the proposed method.

## 2 Related Work

### 2.1 Multi-view Clustering

Numerous MVC methods have been proposed in the past few decades. For example, Large-scale Multi-View Spectral Clustering (MVSC) approximates each view's similarity graph via a bipartite anchor graph, reducing time and space complexity of spectral clustering to near-linear while preserving cross-view manifold structure (27). Another line of work focuses on adaptive fusion: Self-weighted Multi-view Clustering (SwMC) learns view-specific Laplacian graphs and their confidence weights jointly, enabling direct cluster assignment without an extra $k$-means step (10). To further account for sample-specific view importance, Localized multiple kernel $k$-means clustering extends kernel $k$-means to multi-view data by learning sample-specific kernel weights through localized data fusion, which adaptively captures view importance and excels on biomedical datasets (28). Diversity-Induced Multi-View Subspace Clustering (DiMSC) leverages the Hilbert–Schmidt Independence Criterion to enforce statistical independence among view-specific representations and thus enhance complementarity (13). However, these methods rely on shallow, linear representations and often fail to capture complex nonlinear dependencies across views, motivating the shift towards deep multi-view clustering techniques.

### 2.2 Deep Multi-view Clustering

Deep Multi-view Clustering (DMVC) leverages deep neural networks to learn non-linear representations from multi-view data. Early approaches primarily focused on learning a shared representation

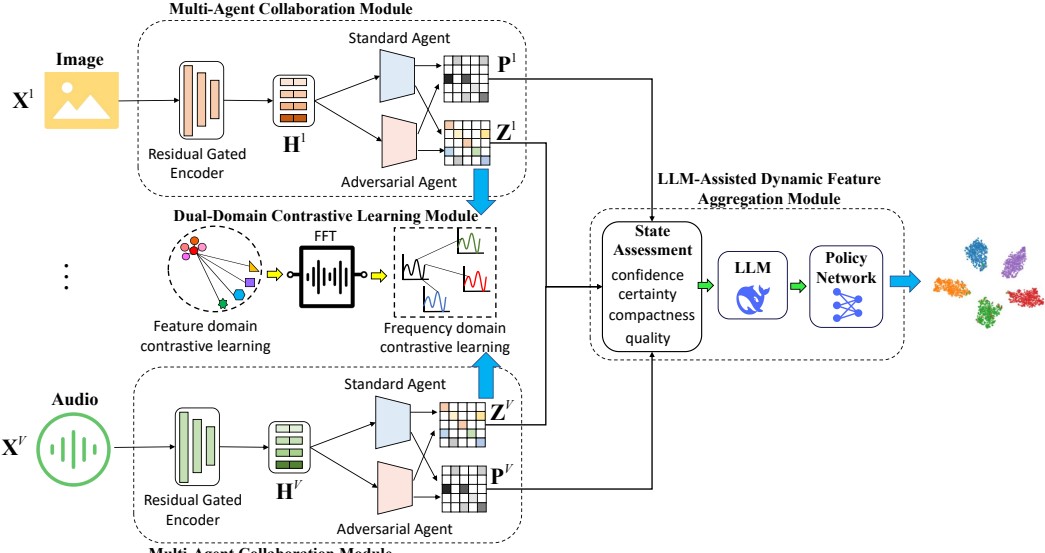

Figure 1: The overall architecture of the proposed LLM-DAMVC framework. Multi-view data $(\mathbf{X}^1, \dots, \mathbf{X}^V)$ are processed by a Residual Gated Encoder to generate shared latent features ($\mathbf{H}^v$). These features are then fed into two parallel branches, a Standard Agent and an Adversarial Agent, to produce initial predictions ($\mathbf{P}_{\text{std}}^v$, $\mathbf{P}_{\text{adv}}^v$) and contrastive embeddings ($\mathbf{Z}_{\text{std}}^v$, $\mathbf{Z}_{\text{adv}}^v$). The encoder is subsequently refined by a Dual-Domain Contrastive Learning Module, which enforces consistency on these embeddings in both the feature domain and frequency domain (via FFT). Concurrently, a state assessment based on multi-faceted quality indicators ($\mathbf{q}_a^v$) informs an **LLM** and a parallel heuristic policy to generate dynamic aggregation weights ($\mathbf{w}_a^v$). Finally, these weights guide the fusion of all agent predictions into a unified probability distribution ($\mathbf{P}$), from which the final cluster assignments are derived.

space, often employing autoencoders for cross-view reconstruction (29) or deep canonical correlation analysis (DCCA) to maximize inter-view correlations (30; 31). Subsequent advances explored more sophisticated architectures. Graph-based methods, for instance, utilize Graph Neural Networks (GNNs) to capture the underlying topological structure of the data (16; 32). More recently, to address the challenge of static view fusion, attention mechanisms have been widely adopted to learn dynamic, instance-specific weights for view aggregation (33; 22). However, while these attention-based methods offer improved adaptability, they typically rely on predefined rules to assign weights for each view, making them inflexible when applied in complex scenarios and processing heterogeneous views.

## 3 METHODOLOGY

### 3.1 Notations and Problem Definition

Multi-view clustering aims to leverage complementary information from different feature spaces to achieve better clustering performance. Given a multi-view dataset $\mathcal{X} = \{\mathbf{X}^1, \mathbf{X}^2, \dots, \mathbf{X}^V\}$, where $\mathbf{X}^v \in \mathbb{R}^{N \times d_v}$ represents the feature matrix of the $v$-th view containing $N$ samples with $d_v$-dimensional features. The goal of multi-view clustering is to partition these $N$ samples into $K$ semantically consistent clusters $\{\mathcal{C}_1, \mathcal{C}_2, \dots, \mathcal{C}_K\}$, where $K$ represents the number of inherent classes in the dataset.

In this section, we introduce a novel deep multi-view clustering method called LLM-assisted Dynamic Agent Multi-View Clustering (LLM-DAMVC), which reformulates MVC as a dynamic decision-making problem. Our approach introduces specialized agents that process individual views and dynamically adjust their contributions based on their performance. Instead of using a predefined fusion strategy, LLM-DAMVC employs a dynamic routing mechanism that determines how to optimally combine these features based on their quality and complementarity. LLM-DAMVC mainly consists of three key modules: Multi-Agent Collaboration Module, Dual-Domain Contrastive

Learning Module, and LLM-Assisted Dynamic Feature Aggregation Module. Specific descriptions of these modules will be provided in the subsequent sections.

## 3.2 Multi-Agent Collaboration Module

The Multi-Agent Collaboration Module serves as the fundamental feature extraction component of our LLM-DAMVC framework. This module is composed of Residual Gated Encoder, Standard Agent, and Adversarial Agent. The Residual Gated Encoder transforms the input into a latent representation $\mathbf{H}^v$. The two specialized agents operate on the latent representation $\mathbf{H}^v$ in parallel for distinct and complementary purposes.

**Residual Gated Encoder:** Given the input data matrix $\mathbf{X}^v \in \mathbb{R}^{N \times d_v}$ for view $v$, we first transform it into a view-specific latent representation through a residual gated encoder $\mathbf{E}_v$. For the first $L-1$ layers, each is composed of a linear layer, a batch normalization (BN) layer, and an activation layer. Starting with the initial input $\mathbf{H}_0^v = \mathbf{X}^v$, each intermediate layer transforms the features as:

$$\mathbf{H}_l^v = \mathrm{ReLU}\left(\mathrm{BN}(\mathbf{H}_{l-1}^v \mathbf{W}_l + \mathbf{b}_l)\right), \quad l = 1, 2, \ldots, L-1 \tag{1}$$

where $\mathrm{ReLU}(\cdot)$ denotes the ReLU activation function; $\mathrm{BN}(\cdot)$ is batch normalization, and $\mathbf{W}_l$, $\mathbf{b}_l$ are learnable parameters. The final layer ($L$-th layer) incorporates a self-gating mechanism and a residual connection to enhance representational capacity (34; 35):

$$\mathbf{H}^v = \left(1 + \mathrm{sigmoid}(\mathbf{H}_{L-1}^v \mathbf{W}_{\mathrm{gate}})\right) \odot \mathbf{H}_{L-1}^v + \alpha f(\mathbf{X}^v) \tag{2}$$

where $\mathbf{H}^v \in \mathbb{R}^{N \times d}$ represents the latent embedding, $\mathrm{sigmoid}(\cdot)$ is the sigmoid function, $\mathbf{W}_{\mathrm{gate}}$ is the gating parameter, $\odot$ denotes element-wise multiplication, and $\alpha > 0$ is a scalar hyperparameter controlling the strength of the residual connection. Crucially, $f(\cdot)$ is a linear projection to ensure dimensional consistency for the residual connection.

**Standard Agent:** We first incorporate a standard agent to extract stable features and learn cluster structure, which transforms the latent representation into a clustering probability distribution and a standard embedding. To be specific, we utilize a learnable mapping $\phi_{\mathrm{std}}$ composed of two linear layers, a BN layer, a ReLU activation layer, and a softmax operation to learn the clustering probability distribution $\mathbf{P}_{\mathrm{std}}^v \in \mathbb{R}^{N \times K}$ from $\mathbf{H}^v$ as follows:

$$\mathbf{P}_{\mathrm{std}}^v = \phi_{\mathrm{std}}(\mathbf{H}^v) \tag{3}$$

Then we employ another learnable mapping $\mathcal{E}_{\mathrm{std}}$ composed of two linear layers, a BN layer, and a ReLU activation layer, to project $\mathbf{H}^v$ into a standard embedding $\mathbf{Z}_{\mathrm{std}}^v \in \mathbb{R}^{N \times d_z}$:

$$\mathbf{Z}_{\mathrm{std}}^v = \mathcal{E}_{\mathrm{std}}(\mathbf{H}^v) \tag{4}$$

**Adversarial Agent:** Operating in parallel, we introduce an adversary agent that explores the boundaries of the feature space to enhance robustness and discriminability. It first learns the adversary clustering probability distribution $\mathbf{P}_{\mathrm{adv}}^v$ through $\phi_{\mathrm{adv}}$ that has the same structure as $\phi_{\mathrm{std}}$:

$$\mathbf{P}_{\mathrm{adv}}^v = \phi_{\mathrm{adv}}(\mathbf{H}^v) \tag{5}$$

To obtain the adversarial embedding $\mathbf{Z}_{\mathrm{adv}}^v \in \mathbb{R}^{N \times d_z}$, we use a mapping $\mathcal{E}_{\mathrm{adv}}$ with similar structure to $\mathcal{E}_{\mathrm{std}}$:

$$\mathbf{Z}_{\mathrm{adv}}^v = \mathcal{E}_{\mathrm{adv}}(\mathbf{H}^v) \tag{6}$$

Then, we compute $\hat{\mathbf{y}}^v = \arg\max(\mathbf{P}_{\mathrm{adv}}^v)$ as pseudo-labels, which guides the generation of adversarial samples $\tilde{\mathbf{X}}^v$ by computing input gradients:

$$\tilde{\mathbf{X}}^v = \mathbf{X}^v + \delta \cdot \mathrm{sign}\left(\nabla_{\mathbf{X}^v} \mathrm{CE}(\mathbf{P}_{\mathrm{adv}}^v, \hat{\mathbf{y}}^v)\right) \tag{7}$$

where $\nabla$ represents gradient operation, CE denotes the cross-entropy operation, $\mathrm{sign}$ is the sign function, and $\delta > 0$ controls the perturbation intensity. Then, $\tilde{\mathbf{X}}^v$ is used to retrain the adversarial agent, enforcing robustness against input perturbations, and is not propagated to subsequent modules. Subsequently, we build a discriminator $\psi$ with MLP to distinguish real features ($\mathbf{H}^v = \mathbf{E}_v(\mathbf{X}^v)$) and

generated features ($\tilde{\mathbf{H}}^v = \mathbf{E}_v(\tilde{\mathbf{X}}^v)$). This adversarial branch introduces a min-max game between the encoder and the discriminator to enhance representation robustness with discriminator loss $\mathcal{L}_{disc}$.

$$\mathcal{L}_{disc} = -\mathbb{E}_{\mathbf{X}^v}[\log(\psi(\mathbf{H}^v))] - \mathbb{E}_{\tilde{\mathbf{X}}^v}[\log(1 - (\psi(\tilde{\mathbf{H}}^v)))] \tag{8}$$

We compute discrimination scores by $\mathbf{s}_{adv}^v = \psi(\mathbf{H}^v)$ ($\mathbf{s}_{adv}^v \in \mathbb{R}^{N \times 1}$). To enforce cross-view alignment and high-quality clustering, we add an alignment loss as follows:

$$\mathcal{L}_{align} = -\sum_{v=1}^{V} \sum_{u \neq v} \mathcal{A}(\mathbf{H}^v, \mathbf{H}^u) + \frac{1}{V(V-1)} \sum_{v=1}^{V} \sum_{u \neq v} \mathcal{D}_{KL}\left(\mathbf{P}_{std}^v \,\|\, \mathbf{P}_{std}^u\right) \tag{9}$$

where $\mathcal{A}(\cdot, \cdot)$ maximizes the canonical correlation between view-specific latent features $\mathbf{H}^v$ by operating on their cross-covariance matrix, and $\mathcal{D}_{KL}$ is KL-divergence to ensure cluster distribution consistency.

### 3.3 Dual-Domain Contrastive Learning Module

To improve the discriminativeness and consistency, we propose a Dual-domain Contrastive Learning module that integrates analyses of the feature domain and the frequency domain. This module uses the standard embedding $\mathbf{Z}_{std}^v$ and the adversary embedding $\mathbf{Z}_{adv}^v$ to build contrastive loss in the two domains. For easier presentation, we employ $\mathbf{z}_i^v$ to represent the embedding of $i$-th sample in view $v$ from $\mathbf{Z}_{std}^v$ or $\mathbf{Z}_{adv}^v$.

**Feature-domain Contrastive Learning:** For corresponding samples across different views $v$ and $u$, we compute their cosine similarity:

$$s_{ij}^{v,u} = \frac{(\mathbf{z}_i^v)^\top \mathbf{z}_j^u}{\|\mathbf{z}_i^v\| \cdot \|\mathbf{z}_j^u\|} \tag{10}$$

Following the contrastive learning framework (36), we consider features from different views of the same sample $i$ as positive samples and a different sample $j$ ($j \neq i$) as negative samples. Then, we construct the contrastive loss as follows:

$$\mathcal{L}_{feature} = -\frac{1}{N} \sum_{1 \leq v \leq V, u \neq v} \sum_{1 \leq i \leq N} \log \frac{\exp(s_{ii}^{v,u}/\tau)}{\sum_{j=1}^{N} \exp(s_{ij}^{v,u}/\tau)} \tag{11}$$

where $\tau > 0$ is a temperature hyperparameter. This objective promotes alignment of corresponding cross-view samples while enforcing separation from non-corresponding instances.

**Frequency-domain Contrastive Learning:** We extend contrastive learning to the frequency domain to capture global structural patterns that remain elusive in the feature space. For each feature vector $\mathbf{z}_i^v \in \mathbb{R}^{d_z}$, we apply the Fast Fourier Transform (FFT) to compute the frequency-domain feature $\hat{\mathbf{z}}_i^v = \mathcal{F}(\mathbf{z}_i^v)$, where $\mathcal{F}$ denotes the FFT operation and $\hat{\mathbf{z}}_i^v \in \mathbb{C}^{d_z}$ is the complex-valued frequency spectrum. We extract the amplitude spectrum $|\hat{\mathbf{z}}_i^v| \in \mathbb{R}^{d_z}$, which characterizes the energy distribution across frequency components while maintaining translation invariance.

For the embedding $\mathbf{z}_i^v$ of an anchor sample $i$ from view $v$, we construct a contrastive triplet $(\mathbf{z}_i^v, \mathbf{z}_i^u, \mathbf{z}_j^u)$ by selecting corresponding embedding $\mathbf{z}_i^u$ in another view $u$ ($u \neq v$) as positive sample and select another sample $\mathbf{z}_j^u$ ($j \neq i$) as negative sample. For efficient computation, we only sample $M$ negative samples, and the frequency-domain contrastive loss is defined as follows:

$$\mathcal{L}_{frequency} = \frac{1}{NMV} \sum_{1 \leq v \leq V, u \neq v} \sum_{1 \leq i \leq N} \sum_{j \in \mathcal{J}(i)} \frac{\||\hat{\mathbf{z}}_i^v| - |\hat{\mathbf{z}}_i^u|\|_1}{\||\hat{\mathbf{z}}_i^v| - |\hat{\mathbf{z}}_j^u|\|_1 + \epsilon} \tag{12}$$

where $\epsilon > 0$ is a small constant for numerical stability, $\mathcal{J}(i)$ represents the set of indices for negative samples corresponding to anchor $i$.

We obtain the dual-domain contrastive loss with a balancing factor $\rho$ as follows:

$$\mathcal{L}_{cont} = \mathcal{L}_{feature} + \rho \mathcal{L}_{frequency}, \tag{13}$$

The dual-domain contrastive learning enhances the features and helps to capture both local discriminative details and global structural patterns.

## 3.4 LLM-Assisted Dynamic Feature Aggregation Module

To address the limitation of static fusion strategies in traditional MVC, we propose the LLM-Assisted Dynamic Feature Aggregation Module, which leverages an LLM to dynamically assess and aggregate multi-view representations. This module processes the latent features $\{\mathbf{H}^v\}_{v=1}^V$, clustering predictions $\{\mathbf{P}_{\text{std}}^v, \mathbf{P}_{\text{adv}}^v\}_{v=1}^V$, and discriminative score $\{\mathbf{s}_{\text{adv}}^v\}_{v=1}^V$ from the preceding stages.

For an agent of the $v$-th view with type $a$ ($a \in \{\text{adv}, \text{std}\}$), we compute a quality indicator vector $\mathbf{q}_a^v \in \mathbb{R}^{4 \times 1}$ to provide a multi-faceted assessment of its real-time performance:

$$\mathbf{q}_a^v = \begin{bmatrix} \mathbb{E}[\max(\mathbf{P}_a^v(i,:))] \\ \zeta(\mathbf{H}^v, \mathbf{P}_a^v) \\ 1 - \mathcal{H}(\mathbf{P}_a^v) \\ \mathbb{E}[\mathbf{s}_{\text{adv}}^v(i)] \end{bmatrix} \tag{14}$$

where $\mathbb{E}[\max(\mathbf{P}_a^v(i,:))]$ measure the agent's average prediction confidence, $\zeta(\mathbf{H}^v, \mathbf{P}_a^v)$ measures cluster structure quality, $\zeta$ calculates a compactness score based on intra-cluster distances, $1 - \mathcal{H}(\mathbf{P}_a^v)$ measures prediction certainty, $\mathcal{H}$ is entropy operation, and $\mathbb{E}[\mathbf{s}_{\text{adv}}^v(i)]$ measures the average representation quality score, respectively.

The LLM-assisted mechanism processes these quality indicators $\mathbf{q}_a^v$ to generate a dynamic aggregation strategy. Specifically, the quality vectors are formatted into a structured prompt and fed to a frozen large language model (LLM). The LLM analyzes the global performance of all agents and outputs a high-level routing policy, the core component of which is a confidence threshold $\tau_c$ for filtering out unreliable agents. Based on this policy, we compute adaptive aggregation weights for each agent. The raw weight $r_a^v$ for each agent is determined by:

$$r_a^v = \begin{cases} \mathbb{E}[\max(\mathbf{P}_a^v(i,:))] \cdot \|\mathbf{H}^v\|_F, & \text{if } \mathbb{E}[\max(\mathbf{P}_a^v[i,:])] \geq \tau_c \\ 0, & \text{otherwise} \end{cases} \tag{15}$$

where $\|\cdot\|_F$ denotes the Frobenius norm. The raw weights are then normalized to sum to one via softmax, yielding the final weight $\{w_a^v\}$ as follows:

$$w_a^v = \frac{\exp(r_a^v)}{\sum_{v=1}^V \sum_{a \in \{\text{std,adv}\}} \exp(r_a^v)}. \tag{16}$$

The final aggregated prediction is obtained by weighted fusion of all agent predictions:

$$\mathbf{P} = \sum_{v=1}^V \sum_{a \in \{\text{std,adv}\}} w_a^v \mathbf{P}_a^v \in \mathbb{R}^{N \times K}. \tag{17}$$

The clustering loss is formulated as:

$$\mathcal{L}_{\text{clus}} = -\sum_{i \in \mathcal{I}_{\text{high}}} \sum_{k=1}^K y_{ik} \log \mathbf{P}_{ik} + \lambda_1 \sum_{k=1}^K \left| \bar{\mathbf{P}}_k - \frac{1}{K} \right| + \lambda_2 \sum_{i=1}^N (-\log \max(\mathbf{P}_i)) \tag{18}$$

where $\mathbf{P}_{ik}$ is the $(i,k)$-th entry of the aggregated prediction $\mathbf{P}$; $\mathcal{I}_{\text{high}}$ is the set of high-confidence samples, and a sample $i$ is considered as a high-confidence sample if $\max(\mathbf{P}_i)$ exceeds a predefined threshold; $\mathbf{y}_i$ is one-hot vector indicating the assignment of sample $i \in \mathcal{I}_{\text{high}}$, and $y_{ik} = 1$ for $\text{Ind}(\max(\mathbf{P}_i))$ and 0 otherwise; $\text{Ind}(\cdot)$ is the indexing operation. The second term encourages balanced cluster sizes by minimizing the L1 deviation of the mean assignment $\bar{\mathbf{P}}_k = \frac{1}{N} \sum_i \mathbf{P}_{ik}$ from the uniform distribution $1/K$, and the third term promotes prediction clarity.

## 3.5 The Objective Function

The total loss function combines the losses above and is as follows:

$$\mathcal{L}_{\text{total}} = \mathcal{L}_{disc} + \gamma \mathcal{L}_{align} + \lambda \mathcal{L}_{\text{cont}} + \beta \mathcal{L}_{\text{clus}} \tag{19}$$

where $\gamma$, $\lambda$, and $\beta$ are hyperparameters balancing different parts of the objective.

The final cluster assignments are directly obtained from the aggregated prediction of the LLM-Assisted Dynamic Feature Aggregation:

$$c_i = \arg\max_k (\mathbf{P}_{ik}) \quad \text{for } 1 \leq i \leq N, \tag{20}$$

where $c_i \in \{1, 2, \ldots, K\}$ denotes the cluster index for the $i$-th sample.

# 4 EXPERIMENT

## 4.1 Datasets & Metric

We evaluate our method on six benchmark multi-view datasets to validate its effectiveness across diverse data types: **NUS-WIDE (NUS)** (37), an image dataset containing 6,251 samples described by 5 visual feature views; **BDGP** (38), a Drosophila image dataset with 2,500 samples and 2 views (visual and textual); **Handwritten** (39), a digit recognition dataset comprising 2,000 samples represented by 6 heterogeneous feature views; **MNIST-USPS** (38), a cross-domain digit benchmark with 5,000 samples from two complementary image datasets; **Reuters** (40), a multilingual news corpus consisting of 1,800 short articles and their associated topics, represented by 5 language-specific views; and **CCV** (40), a consumer video dataset with 6,773 samples and 3 deep feature views. We report three standard clustering metrics: Accuracy (ACC), Normalized Mutual Information (NMI), and Purity (PUR). All experiments are implemented in PyTorch and conducted on an NVIDIA GeForce RTX 4090 GPU.

## 4.2 Comparing Methods

To evaluate LLM-DAMVC on complete multi-view datasets, we benchmark it against state-of-the-art multi-view clustering (MVC) methods, spanning classical baselines, deep representation learning approaches, contrastive learning techniques, and advanced dynamic fusion strategies. As a classical baseline, **K-Means** (41) minimizes intra-cluster distances for partitioning but overlooks multi-view complementarity. Deep MVC methods include **DCP** (42), **EE-IMVC** (43), **ASR** (44), and **MFLVC** (24). Contrastive approaches include **CVCL** (18) and **COMPLETER** (25). Advanced techniques include **ADMC** (45), which enhances efficiency with active sample selection in semi-supervised settings, and **COPER** (40), which aligns cluster assignments through correlation-based permutations.

Table 1: Clustering performance comparisons on 6 selected datasets. Best results in **blue bold**, second best in **red bold**.

| Method | NUS | | | BDGP | | | Handwritten | | | MNIST-USPS | | | Reuters | | | CCV | | |
|---|---|---|---|---|---|---|---|---|---|---|---|---|---|---|---|---|---|---|
| | ACC | NMI | PUR | ACC | NMI | PUR | ACC | NMI | PUR | ACC | NMI | PUR | ACC | NMI | PUR | ACC | NMI | PUR |
| **K-Means** (41) | 20.90 | 15.45 | 32.19 | 45.72 | 27.43 | 46.20 | 38.95 | 37.21 | 39.45 | 49.30 | 45.40 | 52.90 | 6.70 | 5.90 | 4.01 | 5.12 | 8.97 | 5.01 |
| **EE-IMVC** (43) | 22.29 | 10.35 | 39.02 | 88.00 | 71.76 | 87.76 | 89.30 | 81.07 | 89.30 | 76.00 | 68.04 | 76.48 | 19.05 | 11.39 | 16.38 | 23.37 | 18.22 | 26.46 |
| **ASR** (44) | 21.50 | 13.73 | 38.97 | 97.68 | 92.63 | 97.68 | 93.95 | 88.26 | 93.95 | 97.90 | 94.72 | 7.90 | 15.51 | 18.42 | 14.55 | 24.06 | 17.41 | 22.73 |
| **DSIMVC** (46) | 28.89 | 18.06 | 43.27 | 99.04 | 96.86 | 99.04 | 87.20 | 80.39 | 87.20 | 99.34 | 98.13 | 99.34 | 24.35 | 22.17 | 20.41 | 31.90 | 30.70 | 30.54 |
| **DCP** (42) | 19.43 | 5.45 | 30.72 | 97.04 | 92.43 | 97.04 | 85.75 | 85.05 | 85.75 | 99.02 | 97.29 | 99.02 | 26.66 | 32.74 | 21.04 | 20.04 | 16.61 | 14.22 |
| **MFLVC** (24) | 24.11 | 16.43 | 30.87 | 98.72 | 96.13 | 98.72 | 86.55 | 85.98 | 86.55 | 99.66 | 99.01 | 99.66 | 21.14 | 19.98 | 20.37 | 31.23 | 31.60 | 33.90 |
| **CVCL** (18) | 28.65 | 13.96 | 39.83 | 99.20 | 97.29 | 99.20 | 97.35 | 94.05 | 97.35 | 99.70 | 99.13 | 99.70 | 55.64 | 31.14 | 57.35 | 26.23 | 26.25 | 21.17 |
| **COMPLETER** (25) | 22.48 | 7.71 | 38.73 | 59.95 | 56.18 | 59.95 | 69.36 | 73.93 | 69.70 | 89.08 | 88.86 | 89.02 | 25.63 | 31.73 | 25.34 | 24.58 | 22.51 | 21.14 |
| **ADMC** (45) | 24.36 | 15.97 | 40.06 | 96.96 | 96.12 | 96.96 | 84.10 | 81.17 | 84.10 | 91.26 | 95.50 | 91.26 | 77.33 | 70.49 | 77.33 | 23.25 | 20.68 | 23.57 |
| **COPER** (40) | 25.55 | 11.67 | 35.54 | 89.65 | 73.92 | 75.96 | 87.55 | 77.15 | 89.05 | 99.88 | 99.64 | 99.88 | 53.15 | 31.10 | 51.79 | 28.06 | 26.32 | 24.57 |
| **Our Method** | 37.11 | 37.51 | 48.53 | 99.67 | 98.76 | 99.64 | 98.66 | 95.36 | 98.43 | 99.92 | 99.63 | 99.92 | 78.76 | 76.48 | 79.83 | 46.55 | 55.77 | 49.81 |

## 4.3 Experimental Analysis

**Performance Comparison:** To comprehensively evaluate the efficacy of our proposed method, we conduct an extensive comparative analysis against multi-view clustering methodologies across six challenging benchmark datasets. As detailed in Table 1, the empirical results unequivocally demonstrate the superior performance of our approach. Our method, LLM-DAMVC, consistently outperforms all baseline models across all datasets and evaluation metrics, which underscores its exceptional generalizability and robustness to diverse data characteristics. The superiority of our method is particularly illustrated on well-structured datasets such as BDGP and MNIST-USPS, where it achieves superior clustering outcomes with both accuracy and purity levels surpassing 99%. On these datasets, our approach markedly surpasses even the most competitive baselines, showcasing its capability to discern fine-grained cluster structures. Furthermore, when confronted with more intricate and noisy datasets like CCV and Reuters, our method sustains its high efficacy. This consistent performance across varied data modalities validates the effectiveness of our LLM-assisted dynamic fusion mechanism in adaptively handling complex, real-world data distributions.

**Hyper-parameter Analysis:** We analyze the sensitivity of our model to the hyperparameters $\lambda$, $\beta$, and $\gamma$ that weight the contrastive loss ($\mathcal{L}_{\text{cont}}$), clustering loss ($\mathcal{L}_{\text{clus}}$), and alignment loss ($\mathcal{L}_{\text{align}}$). The model is robust to the choice of $\lambda$: performance remains stable over a wide range of values, which

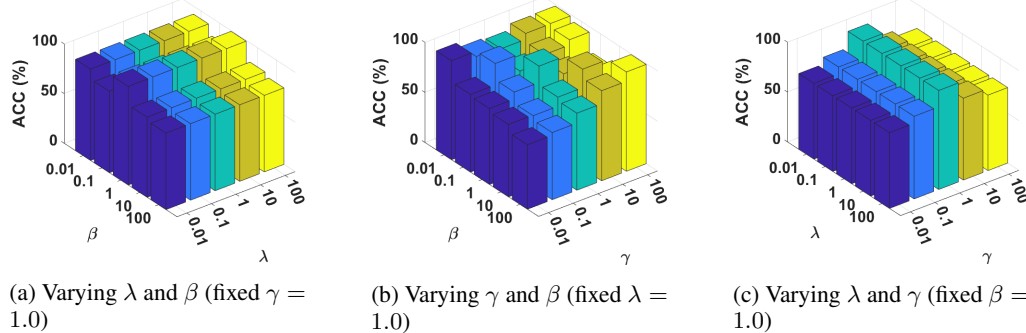

(a) Varying $\lambda$ and $\beta$ (fixed $\gamma = 1.0$)

(b) Varying $\gamma$ and $\beta$ (fixed $\lambda = 1.0$)

(c) Varying $\lambda$ and $\gamma$ (fixed $\beta = 1.0$)

Figure 2: Parameter sensitivity analysis of our method on the MNIST-USPS dataset. We investigate the impact of hyperparameters $\lambda$, $\beta$, and $\gamma$ on clustering accuracy (ACC). Each subplot shows the performance landscape by varying two parameters while keeping the third fixed at its optimal value (e.g., $\gamma = 1.0$).

shows that the contrastive loss consistently fulfills its role of pulling similar samples together and pushing dissimilar ones apart, and this function is reliably effective without requiring precise tuning. In contrast, for the parameters $\gamma$ and $\beta$, there exist performance peaks only at specific values and drops when deviating, demonstrating that the alignment loss and clustering loss must be carefully balanced to work properly. The alignment loss is responsible for establishing cross-view consistency by aligning representations from different views into a shared space, and the clustering loss directly optimizes the cluster structure by encouraging samples to form coherent groups. By comprehensively analyzing the three figures, we found that high performance is achieved when all three losses are present. Removing or severely weakening any one of them leads to degradation, which confirms that each loss performs a distinct and necessary function: contrastive learning handles fine-grained sample discrimination, alignment ensures cross-view agreement, and clustering enforces global semantic grouping.

Table 2: Progressive ablation study showing the contribution of each component in LLM-DAMVC. Best results in **blue bold**, second best in **red bold**.

| Method | NUS | | | BDGP | | | Handwritten | | | MNIST-USPS | | | Reuters | | | CCV | | |
|---|---|---|---|---|---|---|---|---|---|---|---|---|---|---|---|---|---|---|
| | ACC | NMI | PUR | ACC | NMI | PUR | ACC | NMI | PUR | ACC | NMI | PUR | ACC | NMI | PUR | ACC | NMI | PUR |
| Baseline (Only $\mathcal{L}_{\text{feature}}$) | 27.42 | 6.20 | 30.05 | 34.41 | 16.14 | 35.23 | 29.30 | 21.87 | 29.30 | 23.70 | 14.24 | 24.67 | 37.83 | 29.76 | 39.17 | 18.86 | 20.93 | 19.15 |
| + $\mathcal{L}_{\text{disc}}$ | 29.15 | 9.80 | 32.40 | 35.71 | 20.14 | 37.16 | 31.20 | 24.50 | 32.10 | 25.80 | 17.30 | 27.20 | 40.20 | 33.10 | 41.50 | 20.70 | 23.80 | 21.90 |
| + $\mathcal{L}_{\text{frequency}}$ | 31.60 | 14.20 | 36.80 | 40.15 | 26.61 | 40.15 | 55.40 | 48.71 | 55.40 | 65.51 | 62.17 | 65.51 | 45.65 | 38.93 | 47.20 | 23.92 | 28.44 | 25.78 |
| + $\mathcal{L}_{\text{clus}}$ | 34.37 | 20.44 | 40.67 | 81.73 | 78.15 | 81.73 | 87.75 | 90.30 | 87.75 | 97.30 | 97.73 | 98.30 | 72.82 | 65.48 | 73.20 | 43.61 | 51.14 | 45.53 |
| + $\mathcal{L}_{\text{align}}$ (Full) | 37.11 | 37.51 | 48.53 | 99.67 | 98.76 | 99.64 | 98.66 | 95.36 | 98.43 | 99.92 | 99.63 | 99.92 | 78.76 | 76.48 | 79.83 | 46.55 | 55.77 | 49.81 |

**Ablation Studies:** To rigorously evaluate the contribution of each proposed component, we conduct a progressive ablation study, with detailed results presented in Table 2. The findings clearly demonstrate that each module synergistically contributes to the final performance of LLM-DAMVC. Our analysis begins with a baseline model trained solely on the feature-domain contrastive loss ($\mathcal{L}_{\text{feature}}$), which establishes a foundational performance level. From this baseline, we observe a clear and consistent trend of improvement as each new component is incrementally introduced. Notably, the most significant performance increase is observed upon the integration of the frequency-domain contrastive loss (+ $\mathcal{L}_{\text{frequency}}$) and the clustering loss (+ $\mathcal{L}_{\text{clus}}$). The former's impact strongly validates our core hypothesis that incorporating frequency-domain analysis is crucial for capturing global structural patterns that are often missed by conventional feature-domain methods. Meanwhile, the substantial gains from the latter highlight the undeniable necessity of a direct and well-structured clustering objective to guide the model towards forming high-quality and coherent cluster structures. Finally, the inclusion of the discriminator loss (+ $\mathcal{L}_{\text{disc}}$) and the alignment loss (+ $\mathcal{L}_{\text{align}}$) provides further valuable refinements, incrementally improving the model's capabilities to its best performance. The full model, incorporating all components, consistently achieves the best results across all datasets.

**Multi-Dimension Analysis:** Figure 3 illustrates feature representation evolution on BDGP dataset, from unstructured raw features (Figure 3a) to well-defined clustering structures, validating our LLM-DAMVC framework. Figure 4 (left) shows convergence trajectories stabilizing after 60 iterations with monotonic loss descent, demonstrating robust convergence. Figure 4 (right) reveals DeepSeek-8B outperforming across all datasets, particularly on complex ones (NUS, CCV), while even the 1.5B

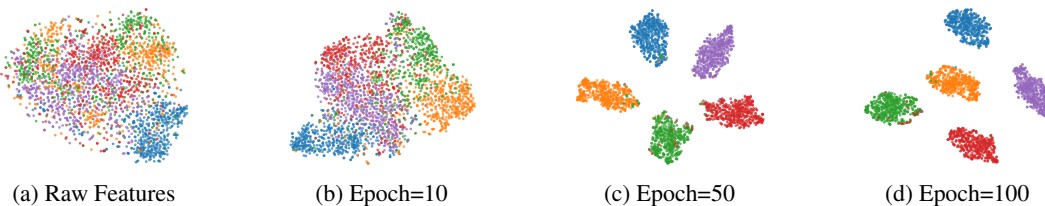

| (a) Raw Features | (b) Epoch=10 | (c) Epoch=50 | (d) Epoch=100 |

Figure 3: T-SNE visualization on the BDGP dataset with increasing training iteration.

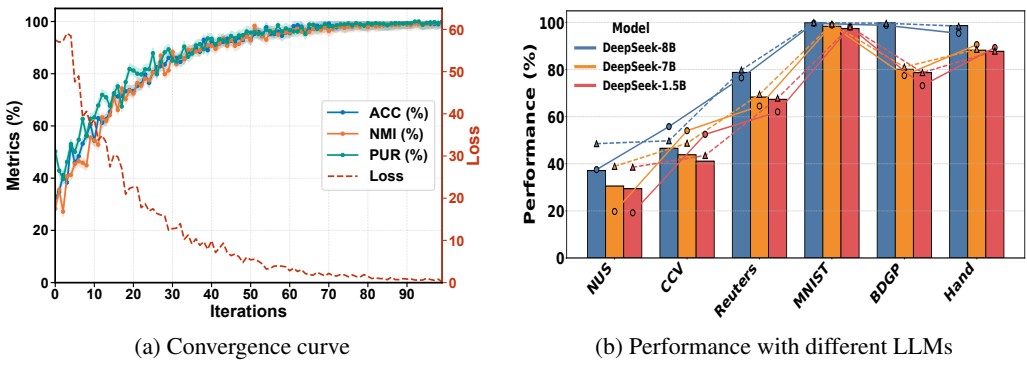

(a) Convergence curve      (b) Performance with different LLMs

Figure 4: Convergence curve and performance of different LLM decision models

variant maintains acceptable performance on structured datasets (MNIST, BDGP). These findings highlight both LLMs' critical role in multi-view clustering and our framework's scalability under diverse computational constraints.

## 4.4 Limitations

Despite our achievements, the designed LLM-Assisted Dynamic Feature Aggregation Module adopts general LLMs to build the decision-making module for view fusion. Though it indicates effectiveness, the performance could be improved by adjusting the LLMs to be dedicated to view fusion tasks in MVC. Therefore, our future work will fine-tune the LLMs to make them more suitable for MVC. Besides, we will also consider introducing the model distillation technique to enhance the efficiency of the dynamic feature aggregation process.

## 5 Conclusions

In this paper, we propose LLM-DAMVC, a novel multi-view clustering framework that reformulates view fusion as a dynamic decision-making process orchestrated by LLMs. To address the limitations of existing methods, *i.e.*, their reliance on feature-domain representation learning and static fusion strategies, we introduce two key modules. First, a dual-domain contrastive learning module jointly optimizes feature consistency and cluster separability in both the original feature domain and the frequency domain via FFT, thereby capturing subtle structural patterns that are indistinguishable in the raw feature space. Second, an LLM-assisted dynamic fusion mechanism leverages semantic reasoning to evaluate view quality at the sample level based on semantic feature quality, cluster structure coherence, and intra-cluster compactness and adaptively assigns fusion weights without predefined rules. Unlike prior works that employ LLMs for feature extraction, our framework uniquely positions the LLM as a decision-making module that coordinates multi-view integration in response to data distribution shifts and view heterogeneity. Extensive experiments across six benchmark datasets demonstrate that LLM-DAMVC consistently outperforms state-of-the-art methods, particularly in complex scenarios with heterogeneous views and varying data quality, validating the effectiveness of our dynamic LLM-assisted clustering method.

# 6  Acknowledgments

This work is supported by the National Natural Science Foundation of China under Grant 62176203, the Fundamental Research Funds for the Central Universities (ZYTS25267, QTZX25004), and the Science and Technology Project of Xi'an (Grant 2022JH-JSYF-0009), Open Project of Anhui Provincial Key Laboratory of Multimodal Cognitive Computation, Anhui University (No. MMC202416), Selected Support Project for Scientific and Technological Activities of Returned Overseas Chinese Scholars in Shaanxi Province 2023-02,

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
