# OpenReview forum: "LLM-DAMVC: A Large Language Model Assisted Dynamic Agent for Multi-View Clustering"
_NeurIPS.cc/2025/Conference — NeurIPS 2025 poster_

### Official Review · Reviewer_yCME · 2025-06-20

**Clarity:** 2
**Significance:** 2
**Originality:** 3
**Rating:** 4
**Confidence:** 4

**Summary:**

this work introduces a large-language model guided dynamic agent multi-view clustering. Overall, the integration of a large language model (LLM) into the learning process of multi-view clustering is interesting. However, there are several serious issues in this work.

1. The motivation of this work is not convincing. for many tasks (not only multi-view clustering), they all excessive reliance on feature extraction. From my perspective, the primary objective of multi-view clustering should be to fully exploit the information across multiple views and uncover the underlying correlations between them, rather than focusing solely on feature extraction. Regarding the second motivation, there already exist numerous studies that incorporate dynamic fusion strategies into the multi-view clustering process.
2. The authors claim that "static or semi-static fusion strategies exhibit limited adaptability when facing drastic view-quality variations at the sample level." However, this argument is not sufficiently supported. More concrete examples or experimental evidence should be provided to substantiate this claim.
3. For the view-quality, it should be provided a detailed definition. Actually, the evaluation of view quality is closely related to the clustering methods employed. For example, view A maybe better than view B when using k-means algorithm, while view A maybe worse than view B when using subspace clustering.
4. The considering of exploring information from frequency domain is interesting. The authors claim that "static or semi-static fusion strategies exhibit limited adaptability when facing drastic view-quality variations at the sample level." However, this argument is not sufficiently supported. More concrete examples or experimental evidence should be provided to substantiate this claim.
5. The overall comparison is both insufficient and potentially biased. Detailed descriptions of the datasets used are not provided, and all selected datasets are relatively small-scale. Moreover, several state-of-the-art multi-view clustering methods—such as low-rank tensor-based approaches (e.g., t-SVD-MSC, TLpNM-MSC)—should also be included as baseline methods for fair comparison.
6. Several typos, such as “Contrastive”-->” contrastive” in page 1 line 13, and “v” in Eq. (1).

**Questions:**

Please refer to "Summary"

**Ethical Concerns:**

["NO or VERY MINOR ethics concerns only"]

**Final Justification:**

My concerns have been addressed, and the clarifications provided helped me better understand the contribution of this work. Therefore, I have decided to raise my overall score.

**Limitations:**

Please refer to "Summary"

**Paper Formatting Concerns:**

Please refer to "Summary"

**Quality:**

2

**Strengths And Weaknesses:**

Please refer to "Summary"

---

> ### Author Rebuttal · Authors · 2025-07-31
>
> **Q1: The motivation appears to overemphasize feature extraction, while the primary objective of Multi-view Clustering (MVC) is arguably the exploitation of cross-view information and correlations. A clarification on how the proposed method balances these two aspects is needed.**
>
> **A1:** Our framework is predicated on the principle that robust feature representation is not a substitute for effective cross-view correlations, but rather a prerequisite for enabling such correlations. We acknowledge that the core of MVC lies in leveraging multi-view information and their inherent correlations. However, without high-quality features, even extensive utilization of multi-view information would yield suboptimal outcomes. To strike this balance, we employ a multi-level optimization strategy. At the foundational level, our model aligns views by minimizing cross-view contrastive loss, ensuring semantic consistency at the sample level. To capture deeper structural similarities, we introduce Frequency-Domain Contrastive Regularization , which aligns the global spectral patterns of features, thereby imposing a more profound form of consistency regularization. Finally, at the decision-making layer, a consistency loss based on KL divergence ensures that the final cluster assignments are consistent across all views. Thus, our approach does not merely focus on feature extraction; instead, it establishes a collaborative hierarchy that fully leverages multi-view information across diverse dimensions (i.e., feature space, frequency domain, and decision space). Therefore, we do not over-rely on feature extraction but rather “treat high-quality features as the cornerstone for multi-dimensional and in-depth cross-view information fusion.
>
>
>
> ---
>
> **Q2: Given that dynamic fusion strategies have been explored in prior MVC research, what are the unique contributions and specific novelties of the proposed dynamic fusion mechanism compared to existing approaches?**
>
> **A2:** While existing schemes explore dynamic fusion, most rely on single metrics for “dynamic weighting” or even adaptive weighting, which fundamentally differs from our approach. Our aggregator leverages a self-attention mechanism to enable context-aware, mutually enhancing feature interactions prior to aggregation. Critically, we pioneer introducing LLMs into the decision loop: rather than performing numerical computations, LLMs execute human-like, logic-based strategic reasoning. This post-LLM integration conducts multi-dimensional state assessment—moving beyond single metrics to holistically evaluate complex states encompassing prediction quality, cluster structure, and resource efficiency. This strategy indicate more flexible capacity when processing complex scenarios (e.g., “Agent A has high confidence but poor cluster balance, while Agent B shows the opposite”) and generate complete strategies featuring primary/backup agents, confidence thresholds, and fusion modes. In summary, compared with existing dynamic fusion strategies, our method is a high-level semantic reasoning-guided fusion strategy.
>
> ---
>
> **Q3: Could a precise, operational definition of "view-quality" be provided? Furthermore, what concrete experimental evidence can be offered to substantiate the central claim that static fusion strategies are inadequate for handling sample-level variations in view quality?**
>
> **A3:** We consider view quality is not a global attribute but a dynamic, sample-specific, and context-dependent state. Therefore, view quality should also be discussed at the sample level rather than solely at the global view level. Our framework operationalizes this insight by forgoing predefined criteria and instead implementing a real-time, multi-dimensional state assessment mechanism: as detailed in our paper, this mechanism evaluates a metric vector for each view on individual samples, encompassing confidence, entropy, cluster balance, feature quality (computed via intra-class distances), resource efficiency (integrating computation time and memory usage), and consistency with historical predictions—making such multi-dimensional evaluation far more comprehensive than static definitions.
>
> To empirically substantiate our claim regarding the inadequacy of static fusion, we conducted a new "View-Quality Perturbation Experiment" on the CCV dataset. We partitioned the data into two class-based subsets (Group A: classes 0-9; Group B: classes 10-19) and selectively introduced significant noise to View 1 for Group A and View 2 for Group B, simulating drastic sample-level quality variations.
>
> | Method                                              | ACC   | NMI   | PUR   |
> | --------------------------------------------------- | ----- | ----- | ----- |
> | Static Fusion (Fixed, Globally-Optimized Weights)   | 21.05 | 24.31 | 23.40 |
> | Semi-Static Fusion (Batch-Level Adaptive Weights)   | 28.71 | 31.54 | 30.22 |
> | Our Dynamic Fusion (Sample-Level LLM-Guided Policy) | 42.92 | 52.06 | 46.20 |
>
> ---
>
> **Q4: The experimental evaluation is not comprehensive enough in two aspects: insufficient description and small scale of the dataset, and the lack of key SOTA baseline methods.**
>
> **A4:** Regarding Dataset Description and Scale,Detailed statistics are provided below. While our selection includes standard benchmarks, we contend that characterizing them as uniformly "small-scale" overlooks their strategic diversity and complexity. Datasets such as NUS-WIDE-10, CCV, and Handwritten are recognized medium-scale benchmarks posing significant challenges due to their high dimensionality, view heterogeneity, and number of clusters.
>
> | Dataset     | Views | Samples | Classes |
> | ----------- | ----- | ------- | ------- |
> | BDGP        | 2     | 2,500   | 5       |
> | MNIST-USPS  | 2     | 5,000   | 10      |
> | Handwritten | 6     | 5,000   | 10      |
> | NUS-WIDE-10 | 5     | 6,251   | 10      |
> | CCV         | 3     | 6,773   | 20      |
> | Reuters     | 5     | 600     | 6       |
>
> We conducted the following comparative experiments using the relevant methods mentioned by the reviewer, including comparisons with t-SVD-MSC and TL-MSC schemes. The results are as follows:
>
> | Dataset         | Method       | ACC       | NMI       | Purity    |
> | --------------- | ------------ | --------- | --------- | --------- |
> | **REUTERS**     | T-SVD-MSC[1] | 27.50     | 10.97     | 28.00     |
> |                 | MVSC-TLRR[2] | 21.09     | 11.72     | 21.61     |
> |                 | **Our**      | **78.76** | **76.48** | **79.83** |
> | **HANDWRITTEN** | T-SVD-MSC[1] | 95.75     | 91.86     | 95.75     |
> |                 | MVSC-TLRR[2] | 81.51     | 79.12     | 81.51     |
> |                 | **Our**      | **98.66** | **95.36** | **98.43** |
> | **BDGP**        | T-SVD-MSC[1] | 93.96     | 83.56     | 93.96     |
> |                 | MVSC-TLRR[2] | 95.32     | 88.61     | 95.32     |
> |                 | **Our**      | **99.87** | **98.76** | **99.64** |
> | **MNITS-USPS**  | T-SVD-MSC[1] | 62.66     | 59.86     | 64.44     |
> |                 | MVSC-TLRR[2] | 63.72     | 58.52     | 67.85     |
> |                 | **Our**      | **99.92** | **99.63** | **99.92** |
>
> [1]Xie Y, Tao D, Zhang W, et al. On unifying multi-view self-representations for clustering by tensor multi-rank minimization[J]. International Journal of Computer Vision, 2018, 126(11): 1157-1179.
>
> [2]Jia Y, Liu H, Hou J, et al. Multi-view spectral clustering tailored tensor low-rank representation[J]. IEEE Transactions on Circuits and Systems for Video Technology, 2021, 31(12): 4784-4797.
>
> **Q5: The manuscript contains several minor typographical and formatting errors that need to be addressed.**
>
> **A5:** We have thoroughly reviewed the entire text, corrected all identified errors, and improved language clarity to ensure the quality of the final version of the article.

---

> > ### Comment · Area_Chair_2DSM · 2025-08-03
> > **Respond to rebuttal and adjust your evaluation**
> >
> > Dear yCME,
> >
> > Thank you for all your work in reviewing so far! You have several concerns regarding some of the claims made by the authors during your initial review. The authors have answered many of these with additional experimental evaluation. Can you please confirm that these sufficiently address your concerns and whether you have any follow-ups? If yes, please do so ASAP so there is enough time for discussion, if necessary.
> >
> > Thank you!
> >
> > AC

---

> > ### Comment · Reviewer_yCME · 2025-08-04
> >
> > Thank you for the detailed response. My concerns have been addressed, and the clarifications provided helped me better understand the contribution of this work. Therefore, I have decided to raise my overall score.

---

### Official Review · Reviewer_MLHN · 2025-06-25

**Clarity:** 3
**Significance:** 4
**Originality:** 3
**Rating:** 5
**Confidence:** 5

**Summary:**

The paper develops a large language model guided dynamic agent multi-view clustering method, which introduces the reasoning capabilities of large-scale language model (LLM) to enhance the performance of the traditional clustering models. Specifically, each view is processed by two complementary agents: a stable feature extraction agent and an adversarial robustness agent, followed by a Dual-Domain Contrastive Learning module that enforces consistency in both the spatial and frequency domains. An LLM-Assisted Dynamic Feature Aggregation (LDFA) module then computes adaptive fusion weights via an LLM-based policy network to integrate view-specific embeddings. The authors also perform several experiments and the corresponding analyses to support the conclusion of the work.

**Questions:**

(1) Have you considered applying more sophisticated clustering algorithms (e.g., k-means or Gaussian mixture models) on \( \mathcal{Z} \) to better exploit its inherent cluster structures?
(2) In clustering, the quality of pseudo labels cannot be guaranteed. Have you considered any countermeasure in the model design or training strategy to mitigate the negative effects of low-quality pseudo labels?

**Ethical Concerns:**

["NO or VERY MINOR ethics concerns only"]

**Final Justification:**

Most of my concerns have been addressed, I will maintain my positive score.

**Limitations:**

Yes

**Paper Formatting Concerns:**

Some charts can be clearer.

**Quality:**

3

**Strengths And Weaknesses:**

Strengths of the paper are:
(1) Reframing multi-view clustering as a dynamic decision-making problem and introducing an LLM as the "decision brain" is a highly innovative concept. It offers a promising solution to the limitations of static or semi-static fusion strategies in traditional MVC methods.
(2) The experimental section of the paper is solid. LLM-DAMVC achieves superior performance on multiple datasets, which supports the authors' claims.
(3) The entire narrative, from problem definition and methodology to experimental analysis, is easy to follow. Figure 1 provides an intuitive visualization of the overall architecture, which helps readers quickly grasp the core ideas of the model.

Weaknesses of the paper are:
(1) Adversarial agents rely on pseudo labels generated by themselves to construct adversarial samples. In the early stages of clustering tasks or when dealing with noisy data, the quality of pseudo labels may be low, which might lead to potential negative influence on performance.
(2) The authors adopt heterogeneous agents for multi-view data, but its justifications and significance are not straightforward.

---

> ### Author Rebuttal · Authors · 2025-07-31
>
> **Q1: How does your framework mitigate the negative impact of potentially unreliable pseudo-labels generated by the adversarial agent, especially in the early stages of training?**
>
> **A1:** We adopt two strategies to address this issue. First, at the loss-function level, we generate pseudo-labels based on cluster centroid proximity, not direct agent output. We then apply a confidence-based filter (high_conf_mask) to select only the most reliable labels for supervision. The influence of this supervision is further controlled via a curriculum learning scheme, where its loss weight (epoch_factor) gradually increases as training stabilizes. Second, our LLM-Assisted Dynamic Feature Aggregation (LDFA) module provides structural-level control. For Equations (17) and (19), the LLM assesses each agent's output quality and assigns dynamic weights. An adversarial agent producing unreliable results receives a lower weight, thus its negative impact on the final fused representation is actively suppressed.
>
> ---
>
> **Q2: Have you considered if applying established clustering algorithms like K-Means on the final embedding$ \mathcal{Z} $ would yield better results than the current direct assignment method?**
>
> **A2:** The entire learning objective, particularly the clustering-specific loss $L_{clus}$ (Eq. 23), is explicitly formulated to train the network to produce a linearly separable embedding space $\mathcal{Z}$. The model is not merely encouraged to group similar samples, but to directly map their final representations to high-confidence categorical distributions. Applying K-Means would introduce a distance-based clustering objective (Euclidean distance minimization) that is not necessarily aligned with the primary, clustering-oriented loss functions used for representation learning. This objective mismatch could lead to suboptimal results by re-interpreting a space that was already optimized for direct assignment. Furthermore, our approach ensures stability of clustering result by avoiding the initialization sensitivity and additional hyperparameters inherent to iterative algorithms like K-Means.
>
> ---
>
> **Q3: Could you clarify the specific justification and significance of using heterogeneous (standard and adversarial) agents, as opposed to a single agent design?**
>
> **A3:** Using heterogeneous agents aims to simultaneously optimize for both stability and discriminability. The Standard Agent is dedicated to learning the stable, core structure of clusters via contrastive learning. Concurrently, the Adversarial Agent acts as a targeted regularizer, exploring the decision boundaries by generating adversarial samples (Eq. 8) to enhance inter-cluster separability and robustness. A single agent would struggle to balance these conflicting objectives. The significance of this dual-agent design is that it yields a final representation that is both structurally coherent and highly discriminative.

---

### Official Review · Reviewer_sGij · 2025-06-25

**Clarity:** 3
**Significance:** 3
**Originality:** 3
**Rating:** 5
**Confidence:** 4

**Summary:**

This paper presents LLM-DAMVC, a novel multi-view clustering (MVC) framework that integrates a Large-scale Language Model (LLM) to enhance the clustering process. LLM-DAMVC introduces three key innovations: a multi-agent collaboration module that extracts more robust and distinctive features using different agents; a dual-domain contrastive learning module that employs Fast Fourier Transform (FFT) to simultaneously capture local details and global structure; and an LLM-guided dynamic feature aggregation mechanism that adaptively assigns fusion weights based on real-time evaluations of view quality.

**Questions:**

1. What are the specific advantages of using a Large Language Model (LLM) for dynamic weight assignment, especially when compared to traditional attention mechanisms or the lightweight policy network discussed in the paper?
2. In what ways does frequency-domain analysis enhance performance for general high-dimensional feature vectors, particularly when the data is not derived from signals?

**Ethical Concerns:**

["NO or VERY MINOR ethics concerns only"]

**Final Justification:**

The authors' response has addressed my concerns. I would like to keep my original score.

**Limitations:**

Yes.

**Quality:**

3

**Strengths And Weaknesses:**

- Strengths
	1. This paper introduces an innovative use of Large Language Models (LLMs) to support multi-view feature fusion in MVC tasks, offering a fresh approach to addressing core challenges in the field.
	2. The overall design is well-structured and technically sound, demonstrating a thoughtful integration of multiple components.
	3. Extensive experiments and comparative studies demonstrate the effectiveness of the proposed method.
- Weaknesses
	1. Incorporating LLMs into MVC may introduce substantial computational overhead, raising questions about the practical necessity of using LLMs in this context.
	2. The dependence of the FFT-based frequency-spectral contrastive learning on specific data types (e.g., texts or signals) needs to be further clarified. Its generalization ability also needs to be discussed.

---

> ### Author Rebuttal · Authors · 2025-07-31
>
> **Q1: What is the unique advantages of using an LLM for dynamic weight assignment compared to traditional attention mechanisms or the lightweight policy network mentioned in the paper?**
>
> **A1:** The unique advantage of the LLM lies in its capacity for semantic reasoning and structured decision-making, which transcends the direct input-to-weight mapping learned by conventional attention or policy networks. As formulated in Equations (18) and (19), the LLM does not just learn a function; it interprets a structured prompt containing semantically meaningful metrics (e.g., prediction confidence, cluster quality). This allows it to perform higher-order reasoning, such as identifying a view as "high-confidence but low-certainty" and dynamically adjusting the fusion strategy accordingly—a capability that simple neural networks struggle to generalize, especially on out-of-distribution or complex view quality combinations. While the policy network provides efficient, fine-grained adjustments, the LLM provides a robust, logic-driven macro-policy. Our ablation study quantitatively confirms that removing this reasoning component leads to a catastrophic performance drop, demonstrating LLM's important contribution.
>
> ---
>
> **Q2: How does frequency-domain analysis help to improve performance for generic high-dimensional feature vectors that are not from signals or images?**
>
> **A2:** For any feature vector, the Fast Fourier Transform (FFT) provides a principled decomposition into components of varying "frequencies," where we conceptualize low-frequency components as representing the global, slowly-varying structure of the feature manifold, and high-frequency components as capturing local, fine-grained details and noise. Our Frequency-Spectral Contrastive Learning (Eq. 13-15) operates on the magnitude spectrum of these features, which is invariant to translation (i.e., shifts in the feature vector). By encouraging the low-frequency spectra of corresponding samples from different views to be similar, our model learns to align their global structural patterns, effectively acting as a powerful structural regularizer. This forces the model to capture consistent high-level semantics across views while simultaneously reducing sensitivity to high-frequency noise or minor feature perturbations.

---

> > ### Comment · Reviewer_sGij · 2025-08-05
> >
> > Thanks for the further clarifications. I would like to keep my original score at this stage.

---

### Official Review · Reviewer_hDjh · 2025-07-01

**Clarity:** 3
**Significance:** 3
**Originality:** 4
**Rating:** 5
**Confidence:** 4

**Summary:**

This paper presents LLM-DAMVC, a novel multi-view clustering framework that deploys heterogeneous agents for each view to perform initial feature extraction. The proposed method incorporates dual-domain contrastive learning, encompassing both the feature domain and the frequency domain. It then leverages large language models (LLMs) for dynamic view-quality assessment and attention-based fusion. Extensive experiments on six benchmark datasets demonstrate significant improvements over existing methods.

**Questions:**

Q1. Current evaluations of "view quality" primarily focus on prediction certainty. Nevertheless, a high-quality view can also be defined by its complementarity to other views and its noise level. Can the evaluation of view quality reflect complementarity or noise level?

Q2. Large language models (LLMs) are invoked at specific intervals, generating relatively static strategies. Could this design fail to capture the batch-level dynamics of data? Why not adopt more frequent LLM-driven decisions to achieve finer-grained adaptation?

**Ethical Concerns:**

["NO or VERY MINOR ethics concerns only"]

**Limitations:**

Yes.

**Paper Formatting Concerns:**

The font in the picture could be made slightly larger and clearer.

**Quality:**

3

**Strengths And Weaknesses:**

Strengths:
S1. This is the first study to integrate large language models (LLMs) as a decision-making module for adaptive view fusion, thereby effectively overcoming the limitations of static strategies.
S2. The dual-domain contrastive module integrates spatial similarity and spectral structural patterns, which contributes to enhancing robustness against noise.
S3. Comprehensive evaluations were conducted on 6 datasets, and the results effectively demonstrate the superiority of the proposed method.

Weaknesses:
W1. LLM has been proposed and applied for a few years, and the authors conduct a wider range of literature review and performs a detailed analysis of its application in clustering, so as to demonstrate the novelty of the work.
W2. The work adopts LLM as decision making module, and does the method requires training or fine tuning of LLM? For another, how about the different LLMs influence the clustering performance should be further analyzed.
W3. There are some typographical errors, which should be corrected to improve overall clarity.

---

> ### Author Rebuttal · Authors · 2025-07-31
>
> **Q1: Regarding the novelty of applying LLMs to clustering, and whether the LLM requires training or fine-tuning.**
>
> **A1:** Unlike prior methods using LLMs for static feature extraction, the novelty of our work lie in that we leverage its advanced reasoning to interpret complex inter-view relationships and adapt fusion strategies, without any LLM training or fine-tuning. This core mechanism, combined with our collaborative dual agent architecture and innovative dual domain (spatial+frequency) contrastive learning, is used for multi view clustering.
>
> ---
>
> **Q2: Regarding the analysis of how different LLMs affect performance.**
>
> **A2:** We test the clustering performace with different LLMs (i.e., DeepSeek-R1(8B) and LLAMA3), as summarized below:
>
> | Dataset | DeepSeek-R1(8B)         | LLAMA3(8B)              |
> | ------- | ----------------------- | ----------------------- |
> | NUS     | 37.11 \| 37.51 \| 48.53 | 31.37 \| 29.91 \| 42.14 |
> | BDGP    | 99.67 \| 98.76 \| 99.64 | 99.02 \| 97.66 \| 99.02 |
> | HW      | 98.66 \| 95.36 \| 99.92 | 85.97 \| 82.21 \| 79.87 |
> | MNIST   | 99.92 \| 99.63 \| 99.92 | 92.91 \| 93.83 \| 92.91 |
> | REUTERS | 78.76 \| 76.48 \| 79.83 | 79.83 \| 75.97 \| 80.17 |
> | CCV     | 46.55 \| 55.77 \| 49.81 | 42.98 \| 48.35 \| 41.97 |
>
> ---
>
> **Q3: Does the "view quality" evaluation capture view complementarity and noise levels, beyond just prediction certainty?**
>
> **A3:**  Our view quality assessment metric focuses on evaluating the downstream impact of view features on the final clustering task, rather than directly quantifying noise or complementarity. Although these two metrics cannot be directly measured, indirect inference remains feasible. For instance, a view severely corrupted by noise struggles to generate clear and compact feature representations, resulting in poorly organized, more dispersed clustering structures with larger intra-cluster distances and consequently lower feature quality scores. This metric indirectly quantifies the harm of noise to the clustering task. Regarding complementarity, highly complementary views facilitate the model in forming clearer and more separable clusters by providing unique discriminative information, corresponding to higher feature quality scores, whereas redundant views yield lower scores.Crucially, this metric is not used in isolation but as part of a holistic system-level assessment by our LLM-guided router. The LLM integrates the metric with other indicators (e.g., prediction confidence, entropy, resource usage) to perform context-aware reasoning. For example, it can recognize that a view with an average performance score but persistent divergence from high-confidence consensus views may provide critical decisive resolving information, thus retaining it to address ambiguous samples. This cooperation between the metric and LLM reasoning is a core strength of our framework.
>
> ---
>
> **Q4: Why not call the LLM more frequently to capture finer, dynamic changes in the data?**
>
> **A4:**  We employ the LLM as a strategic planner that sets the overarching fusion policy based on a stable and holistic view of the data, which does not require batch-level frequency. Frequent LLM calls would introduce prohibitive computational latency and risk destabilizing the training process by overreacting to batch-level noise. To capture finer dynamics without these drawbacks, we employ a lightweight policy network that updates every batch. This creates a highly effective "LLM macro-guidance + network micro-adjustment" framework, where the LLM provides robust strategy while the policy network handles rapid, tactical adaptations, thus achieving both clustering performance and computational efficiency.
>
> ---
>
> **Q5: A detailed introduction should be given on the application of LLM in clustering, and some formatting errors should be corrected to improve overall clarity.**
>
> **A5:**  We will thoroughly revise the manuscript and expand our work on the latest applications of LLM in clustering tasks. In addition, we will carefully proofread the entire paper and correct all formatting and formatting errors.

---

> > ### Comment · Area_Chair_2DSM · 2025-08-03
> > **Please read author response and confirm if your questions are addressed**
> >
> > Dear hDjh,
> >
> > Thank you for all your work in reviewing so far! Please confirm that your concerns regarding whether LLMs should be more dynamically integrated and regarding metrics used for evaluating the quality of multiple views is answered by the authors. Please engage with responses early and adjust your scores as needeed.
> >
> > Thank you!
> >
> > AC

---

### Decision · Program_Chairs · 2025-09-17

**Decision:**

Accept (poster)

**Comment:**

**Summary of Contributions**: The paper proposes a multiview clustering framework through a multi-agent framework. Multiple agents extract features, combined with an aggregator, and finally an LLM-guided dynamic fusion of extracted features for unsupervised learning. Feature learning uses spatial and frequency domain and the LLM dynamically learns a notion of "view quality" and "fusion weights". Empirical evaluation on six benchmark datasets demonstrates strong results.

**Strengths**: Reviewers noted the following strengths:

1. Novelty of integrating LLMs as a decision-maker and fusion agend for multiview clustering, explicitly overcoming known issues of static strategies.
2. Importance of using frequency domain feature learning to improve robustness to noise.
3. Exhaustive empirical evaluation
4. A few reviewers noted that writing quality, and exposition is well done and the paper is easy to follow (including visualizations aka Figure 1).

**Weaknesses** : Reviewers noted the following weaknesses in their initial review:

1. Dynamic aggregation has been proposed before potentially reducing novelty, where use of LLMs is the main contribution
2. Lack of clarity on whether LLMs need to be fine-tuned for this purpose
3. Not all design choices are well justified, such as choice of scores of "view quality", justification of use of frequency domain feature learning
4. Lack of description and comprehensiveness of experimental data.
3. Minor typos
6. Impact of the choice of LLMs not justified
7. Potentially missed baselines

**Rebuttal and discussion**: Authors added additional justification for dynamic aggregation, compared performance across different LLMs, added dataset descriptions in their response, and baselines suggested by reviewers. Reviewers maintained or improved scores based on the rebuttal.

**Final recommendation**: Accept. Assuming all additional empirical evaluation is worked into the final manuscript, typos and corrections incorporated, I see that reviewers' major concerns are resolved.